# Potent Acrylamide Determination in Food Products Using Ion-Selective Electrode Technique

**DOI:** 10.3390/membranes11080645

**Published:** 2021-08-23

**Authors:** Sabry Khalil, Alaa El-Beltagy, Mohamed El-Sharnouby

**Affiliations:** 1Department of Food Nutrition Science, College of Science, Taif University, P.O. Box 11099, Taif 21944, Saudi Arabia; aelbeltagu@Tu.edu.sa; 2Department of Biotechnology, College of Science, Taif University, P.O. Box 11099, Taif 21944, Saudi Arabia; m.sharnouby@tu.edu.sa

**Keywords:** ion-associate complexe, membrane sensor, acrylamide detection, food product samples

## Abstract

A potent selective acrylamide liquid sensor based on the reaction of acrylamide with 2-(5-Bromo-2-pyridylazo)-5-[N-n-Propyl-N-(3-Sulfopropyl) amino] aniline reagent is successfully designed. The characteristics slope (52.33 mV/decade), linearity usable range from 1.0 × 10^−7^–1.0 × 10^−1^ molar, limit of detection (1.6 × 10^−8^) molar, selectivity attitude to several inorganic cations, amino acids and sugars, time of response (8 s), lifetime (four months), pH effect on the electrode potential and the basic validation parameters were studied. The desirable pH applicable range was 3.0–6.5, and the restraint of the developed sensor is independent on this working pH range. The deployed electrode was effectively applied for rapid inexpensive analysis of acrylamide cations in food products with comparison to high-performance liquid chromatographic method and the results were agreeable with each other. The obtained data by the suggested electrode were treated statistically and compared with the various recently published acrylamide sensors.

## 1. Introduction

Scientists concert that acrylamide in food also has the prospective to litigate cancer in humans. We put forward that the consumed amount of acrylamide (AA) is decreased, as protection. Modern feedback displays that acrylamide is formed during the boiling/sterilizing of foods containing starch, like bread and potatoes. It is not intentionally added to foods; it is a natural second product of the cooking method, and always found in our food. Laboratory studies display that acrylamide can cause cancer in humans. The consistency of acrylamide in addressed food is presented by the chemical combination between amino acid asparagine and glucose. In addition to methionine, other amino acids like cysteine, arginine, alanine, aspartic acid, glutamine, threonine and valine have also produced acrylamide [1].

Acrylamide is formed during the boiling/sterilizing of foods containing starch, such as bread and potatoes.

The recent enactment will demand food business workers to introduce a site simple, practical proceedings to handle acrylamide within their food safety administration systems [2,3]. Food quality monitoring and delightfulness are of global interest for both the food industry and consumers, which has put a lot of effort to recognize and consummation gauges to decrease acrylamide levels in food. This comprises developing orientation on strategies to frontier acrylamide construction in a diversity of foods and processes.

Hence, it is needful to improve such a frugal, sensitive, specified and quick system, which can estimate acrylamide content in thermally processed foods, ion-selective membrane sensors as simple, quick, mobile, little and economic tools can be utilized to evaluate metal cations [4,5,6].

Several traditional standard analytical techniques have been deployed to detect acrylamide concentration including chromatography, mass spectrometry and chromatography-mass spectrometry. Liquid chromatography-mass spectrometry was utilized to estimate acrylamide in food [7]. Other outright gas chromatography-mass spectrometry was used to estimate acrylamide apart in food and coffee powder [8,9]. Recently, enzyme-linked immunosorbent assay (ELISA) [10] and capillary electrophoresis [11] have also been utilized for the determination of acrylamide.

Anywise, these techniques have the disadvantage of tedious pretreatment samples, long estimation time, high expenses and high practical adjustment validations. The electrochemical analysis is vastly concerned by most scientists in order for its advantages of being quick, simple and economic, and ability to be utilized to recognize on-site searches [12]. Therefore, electrochemical techniques have been widely utilized in biological [13,14,15,16,17,18,19,20,21,22,23,24] and environmental analysis [25,26,27,28,29,30,31,32].

For the past few years, many types of research are reported as electrochemical sensors for acrylamide detection [1,33,34,35,36,37,38,39,40,41,42,43,44], although most of them just focused on improving the sensitivity, stability and other performance of the electrochemical analysis.

Despite the recently reported methods having high sensibility and providing accurate and precise results, some tedious complications were found in their utilities. Therefore, they are not suitable for routine and on-site analysis. Potentiometric measurements depend on the nominated electrode, are very plain, and display several superior validations, such as fluent sample conditions, quick restraint, very eclectic, expanding linearity range of concentration, simple types of equipment with a quite low limit of detection, completed in viscous, colored, and/or turbid solutions, and being economic. However, recently specified reagents belonged to heterodiazo dyes that perform strongly, and stabilized ion-associate with many active cations were created to estimate them in foodstuff, real environmental samples and pharmaceutical formulations by new, very selective and sensitive spectrophotometric determinations [45,46,47,48].

The cited ligand 2-(5-Bromo-2-Pyridylazo)-5-[N-n-Propyl-N-(3-Sulfopropyl) amino] aniline [BrPPSAA] (Figure 1) is firstly prepared as described before [49]. It is a tridentate ligand, which has super sensitivity beside a very good selectivity coefficient for some cations. It forms an ion-associate complex with some different cations like (Co, Ni, Zn, Cu, Fe) at the optimum pH range of 3.0–4.5 with maximum absorption approximately at 602 nm; the molar absorptivity of the formed complexes is very high. Generally, the employed reagent 5-Br-PPSAA is commercially available at reasonable costs. So, we decided to use its practical utility for the construction of acrylamide membrane sensors.

The present work describes the construction and evaluation of a newly acrylamide membrane electrode. The active components in a matrix polyvinyl chloride (PVC) selective electrode are the acrylamide-reagent ion-associate complex. The characteristic slope, usable and quick restraint for acrylamide cations was displayed by the developed sensor within the linearity range 1.0 × 10^−7^–1.0 × 10^−1^ M, the detection limit (1.6 × 10^−8^) M, the selectivity study toward several cations, the lifespan (120 days), the restraint time (8 s), the pH effect on the electrode potential and the basic validation parameters were studied. The applicable range of pH was 3.0–6.5, and the restraint of the designed sensor is independent of pH in such working range. The electrode is effectively applied to determine the acrylamide concentration in foodstuff. The obtained results by the nominated sensor were treated statistically, and compared with the other different recently reported acrylamide sensors.

## 2. Materials and Methods

### 2.1. Sample Products, Materials and Chemicals

Chlorides of calcium, magnesium and/or copper, ammonium and sodium hydroxide, Polyvinylchloride (PVC), acrylamide99+%, alanine, D-fructose, and TEHP; [tri-(2-ethyl hexyl) phosphate] were Aldrich products. Hydrochloric acid, tetrahydrofuran, and L-cystine from Merck (Kenilworth, NJ, USA). Arginine, glycine and glucose were from Fluka (Buchs, Switzerland).

The cited ligand [BrPPSAA] was purchased from local chemical stores in Egypt. Foodstuff samples containing acrylamide (coffee substitutions, potato chips, crispbread, French fries, cereal based foods and toast) were obtained from local food stores in Egypt and Saudi Arabia.

### 2.2. Preparation of Stock Solutions

Stock solutions of Ca^+2^, Mg^+2^, and Cu^+2^ of 0.1 molar solutions were prepared by weighing and dissolving the calculated quantities of each one in bidistilled H_2_O. Solutions of 10^−7^–10^−1^ molar were prepared by dilution.

### 2.3. Sample Preparation for the Estimation of Acrylamide Cations

Foodstuff samples (coffee substitutions (dry and soluble), potato chips, crispbread, French fries, cereal-based foods and toast) were selected for analysis, Naturally, raw foods are not toxic. The collected samples were brought to the laboratory, then mashed/ground. One gram of each sample was mixed with 100 mL of distilled water, and then the pH was adjusted to 4.5 using 1 M HCl and the mixture stirred at 25 °C for 30 min. applying a magnetic stirrer. The stirred mixture was filtered through Whatman 1, stored in dark bottles at refrigerator for acrylamide determination (within 24 h), and then equivalent quantities (200, 350, 375, 225, 120, 175, and 150 mg of them, respectively) were taken for analysis.

The foodstuff samples were further analyzed by the high-performance liquid chromatography (HPLC) method, described before [50] for quantification, and the obtained results were compared with those of the present method.

### 2.4. Fabrication of the Selective Sensor

The selective membrane fabrication was developed as mentioned earlier [51]. It comprises a column electrode of Teflon mutable and a body filled with a liquid membrane “+Ag/Ag Cl”, an internal reference sensor/1.0 × 10^−2^ mol L^−1^ acrylamide solution, 1.0 × 10^−2^ mol L^−1^ KCl (internal reference solution). The complex {Acrylamide[BrPPSAA]}, plasticizer (TEHP) and the polyvinyl chloride (PVC) were finely grounded, then tetra hydro furan was added as a volatile solvent. To the flat end of polyvinyl chloride tubing an appropriate diameter disk was cut and glued with tetrahydrofuran. The electrode body was completed with 0.001M of the acrylamide solution. The electrode was conditioned by dipping it for 24 h in 0.01 M acrylamide solution and stayed in a comparable solution.

### 2.5. Active Component of Liquid-Sensor Layer

The complex {Acrylamide[BrPPSAA]} is the active constituent is formed by the reaction between the cited ligand [BrPPSAA] and acrylamide which contains a reactive electrophilic double bond and an amide group, which can also react. It exhibits both weak acidic and basic properties. In an acidic medium, the electron withdrawing carboxamide group activates the double bond, which as a consequence reacts readily with nucleophilic regent [BrPPSAA]. The maximum absorption of free (5-Br-PSAA and acrylamide) are at 457 and 273 nm, respectively. The UV-visible absorption spectrum of the acrylamide-5-Br-PSAA ion-associate complex was measured in the range of 300–700 nm at the optimum acidic (pH = 6). A maximum absorbance of the complex is observed approximately at 602 nm where the free reagent has minimum absorbance. 

### 2.6. The Potential Layer Conditioned

The selectivity and sensitivity of the ion-selective electrode are deeply affected by the composition and the nature of additives employed. Therefore, the influence of various membrane compositions on the response characteristics of the fabricated acrylamide membrane electrode were studied to optimize the best composition of the nominated electrode. Plasticizer is the interesting constituent of the membrane sensor and has effects on the mobility, the state of ionophore molecules and dielectric constant of membrane. It did not only enhance the workability of the membranes, but also improved the working concentration range, stability and life span of the electrode. Therefore, by taking constant amount of PVC (0.44 g), by changing the ratio between the compound [Acrylamide( BrPPSAA)] and the plasticizer TEHP as follows: [(0.04 g, 0.52 g), (0.05 g, 0.51 g), (0.06 g, 0.50 g), (0.01 g, 0.55 g), (0.02 g, 0.54 g), (0.03 g, 0.53 g), respectively]. The mixture was blended to provide the ion-selective sensor strata. A Teflon tube with an electrode of Ag/AgCl was completed with the recently conditioned mixture, and then put into a gel by heating at a temperature of 375 K for 25 min. The membrane sensor was immersed for 2 h in 10^−3^ M acrylamide solution after cooling.

### 2.7. Measurements of EM F

An Orion 90-00-01 solution including 0.55 M potassium chloride, 1.5 M potassium nitrate, one ml of 40% formaldehyde and 0.05 M sodium chloride was applied to fill the stabilized bridge of the reference electrode. An Orion 90-02 reference electrode was utilized with a mechanical stirrer to give an accuracy of 0.1 mV at laboratory conditions for recording the EMF of the acrylamide electrode system. 

## 3. Results

The interesting basic analytical validations of the fabricated acrylamide ion-selective sensor were investigated to present its important value in food analysis. The selectivity study, detection limit, the characteristics slope, response time and influence of pH on the sensor’s potential were studied.

### 3.1. Calibration Curves

Figure 2 presented the calibration curve of acrylamide electrode The acrylamide sensor’s specific slope is 52.33 mV/decade, the detection limit is 1.6×10^−8^ M as determined from the intersection of the two extrapolated segments of the calibration graph, and the measuring linearity usable range is 1.0 × 10^−7^–1.0 × 10^−1^ M (with correlation coefficient r^2^ = 0.998). Table 1 presented the analytical validation parameters of the suggested acrylamide sensor.

### 3.2. Interference Study

The selectivity behavior of the acrylamide membrane sensor with reference to interfering metal cations and compounds like amino acids or sugars which compete with the precursors in food to reduce acrylamide formation, based on the inhibition of the intermediate compounds formed in acrylamide formation was examined by the separate solution mode or by the MPM, (matched potential mode in the case of sugars and amino acids), reported before [52] using the equations
log K^pot^ _ij_ = [(E_2_ − E_1_)/S] − (1 + Z_i_/Z_j_) log a_i_(1)
where, ai is the AA activity, E_1_ and E_2_ are the potentials of AA and interfering species, zi and zj are the charges of AA and interfering species, respectively, and S is the slope of the sensor calibration plot. By employing the separate solution mode, at the value of EMF with acrylamide cations concentration 0.001 M and, the potential −160 mV. With respect to the matched potential method is employed for interfering compounds (sugars and amino acids depending on the activity ratio of AA (ai) and interfering compound (aj), M refer to the interfering species, the equation is:(2)K^pot^_AA/M_ = ai/(aj)

### 3.3. Dynamic Response Time of the Acrylamide Sensor

The response time of the nominated ion-selective electrode is a very interesting parameter for analytical applications. In this research, the practical restraint time was registered by changing the acrylamide cations concentration in solution, over a concentration range from 1.0 × 10^−7^–1.0 × 10^−1^ M, and the results are depicted in Figure 3. As presented, the acrylamide sensor reached its equilibrium response over the whole concentration range, in a very short time (8 s). This is possible due to the quick interchange kinetics of forming and decomposing the ion-associate complex of acrylamide cations with the reagent [BrPPSAA] on the test solution-membrane interface.

### 3.4. Influence of pH on the Sensor Potential

The dependence of the electrode potential on pH was examined by recording the sensor potential, where AA is involved in acid-base equilibrium, and thus the concentration of the cationic form of AA changes with pH resulting in changes in the sensor potential.

Some drops of hydrochloric acid or sodium hydroxide were added to the 0.001 M acrylamide cations sample understudying. After each addition, the pH was registered, the electromotive force; and EMF of the AA electrode system/reference sensor was recorded after the stabilization of the electrode restraint. The pH influence on the EMF is presented in Figure 4. Lowering and higher than the working pH range (3.0–6.5),the potential reduces at higher values (−162 at pH 7.0, −171 at pH 8.0, and −190 at pH 10) which is attributed to the hydrolysis of acrylamide cations or the incompleteness of the complexation reaction. At lower pH values, the potential increases (−150 at pH 2.5, and −157 at pH 2.0) because of the sensor responses to hydronium ions and acrylamide cations.

### 3.5. Lifespan of the AA Selective Electrode

The duration-time of the nominated acrylamide electrode was examined by recording the specific slope of the sensor kept at 4 °C. For at least six months the systematic realizations were carried out once a week, in recently conditioned solutions in a regular mode. Repeatable, constant and reproducible measurements were attained after four months. A tenuous decrement was seen in the slope of the electrode by 1.0 mV decade^−1^ from 52.33–51.33 mV/decade accompanied by an increment in the detection limit. Afterward, the slope of the electrode reduced progressively, while the limit of detection is increased from 46.28 to 42.15 mV/decade and 1.3 × 10^−7^ to 2.5 × 10^−6^ M, respectively. This possibly emerges from the nomination of the sensor constituents. Therefore, the lifetime of the electrode is about four months, related to the basis of the obtained data.

### 3.6. Acrylamide Estimation in Food Product Specimen

The estimation of acrylamide cations in food product specimen was examined using the suggested ion-selective sensor to test its analytical practical utilities. The techniques standard additions and the calibration curve were employed. The amounts of acrylamide in the samples were computed from predetermined calibration plots. The resulting data by the suggested mode were in good compatibility with the HPLC reference method for all food product samples understudying.

## 4. Discussion

The optimum, real value of the Nernstian slope is 59.1/n (mV/decade) [53]. It is deliberated from the slope curve between the log concentrations of standard solution (M) with the potential recorded in (EMF/mV). The specific Nernstian slope is a very interesting factor to distinguish selective electrodes that are adequately employed in the estimations. In this research, the Nernstian value was 52.33 mV/decade as presented in Figure 2, which is closely agreed with the ideal value. It means that the potential changes with 52.33 mV/decade if the AA concentration changes with one order of magnitude. This value proves that the eclectic sensor is still workable to apply in acrylamide analysis due to the pliable value of the specific Nernstian slope is 52.33 mV/decade.

Among the various membrane compositions that contains of 0.01 g compound [acrylamide(BrPPSAA)] with a mixture of, 0.55 g TEHP, and 0.44 g PVC has the best behavior of the membrane electrode (the calibration curve (Figure 2)). It showed a good Nernstian response with a slope of 52.33 mV/decade for four replicate measurements in the usable rectilinear usable range of 1.0 × 10^−7^–1.0 × 10^−1^ M with a very low limit of detection (1.6 × 10^−8^ M), as determined from the intersection of the two extrapolated segments of the calibration graph.

In this work, some different foreign metal cations and compounds were investigated. The values summarized in Table 2 revealed that the suggested acrylamide ion-selective electrode is highly selective towards acrylamide cations in the presence of those interfering metal cations, and compounds (sugars and amino acids) under test. As was shown from the resulting data, the investigated interfering metal cations (Ca^2+^, Mg^2+^, and Cu^2+^) may be interfering. Thus, the interference can be eliminated by adding lactic acid before the addition of the cited reagent, and EDTA was added after the formation of the acrylamide ion-associate complex as masking agents in the suggested method, and then the mixture was again heated at 95 °C for 5 min. The selectivity behavior is improved because the formed metal complexes are decomposed by the addition of EDTA. In the case of sugars and amino acids, the high selectivity is in relation to the difference in polarity and lipophilic nature of their molecules relative to acrylamide. The mechanism of selectivity is basically based on the stereospecificity and electrostatic environment, and counts on how much fitting is available between the locations of the lipophilicity sites in two competing species in the bathing solution side and those present in the receptor of the ion-exchanger [54]. None of the understudying compounds had a noticeable effect on the potentiometric responses of the nominated sensor towards acrylamide cations. The surprise of the high selectivity of the suggested electrode over the other tested foreign metal cations and compounds, most probably resulting from the high affinity of the carrier molecules for acrylamide cations.

The suggested sensor was successfully utilized for acrylamide cations estimation in foodstuff samples with no pretreatment extraction as presented in Table 3. The data provided proved the practical validations of the technique as shown by the veracity, and preciseness statistical treatment of data.

Additionally, as can be seen from the provided data (Table 3) that the standard additions and the calibration curve techniques were applied. The treatment of the results illustrated that the technique of the calibration curve is recommended in the acrylamide detection while the standard additions technique is less recommended, the recorded error is not more than 0.82% and 4.85% in the two modes, respectively, which is due to the repeatability and veracity of the process.

The reproducibility of the proposed sensor was studied by a series of five membranes with the same composition and the response of these electrodes to acrylamide cations concentration was registered. The calibration curves were plotted to investigate the repeatability and reproducibility of the developed electrode. A similar electrode was applied during five replicate measurements for the repeatability test of the ion-selective electrode, while the responses of five similar electrodes to acrylamide cations concentration were tested for the reproducibility of the electrode in the concentration range of 1.0 × 10^−7^–1.0 × 10^−1^ M of acrylamide cations solutions. The results reveal that the standard deviation of measurements of 1.0 × 10^−1^ M to 1.0 × 10^−7^ M of acrylamide solution with these five sensors was ±1.55 mV. The lower values of the coefficient of variation (0.38) also presented repeatability, reproducibility and the precision of the acrylamide selective electrode.

The provided results by the fabricated acrylamide sensor were treated statistically, compared to the other various previously published sensors. As can be seen in Table 4, a comparison among the interesting validation parameters of the quantitative estimations of acrylamide cations applying various electrodes listed recently in the literature. This comparison was made to prove whether the suggested sensor provides credible results and be assumptive for acrylamide cations in foodstuff samples. As displayed in Table 4, the suggested electrode offers a comparable linearity usable range (1.0 × 10^−7^–1.0 × 10^−1^ M) which is more precious than the other recently published acrylamide selective electrodes [33,34,35,36,37,38,39]. It has a long lifetime (120 days) compared to the other listed sensors, which all have low limits of detection, the lowest is that stated in this developed research (1.6 × 10^−8^ M). Furthermore, the proposed sensor has many advantages, is easy to construct, economical. Thus, it can be confidential to state that our suggested sensor is practically validated in all senses, with other sensors to estimate acrylamide cations concentration.

No considerable intervention was noticed from the constituents present in the samples under investigation. The calibration curves displayed an excellent linear restraint on an expanded linearity concentration range. Most of the techniques exhibit valuable veracity with regard to the real values, and no considerable difference for either veracity or preciseness was presented.

## 5. Conclusions

A constructed acrylamide selective sensor was developed. The suggested electrode is specified by excellent validation parameters, such as short response time, relatively long lifetime and the specified Nernstian slope. The analytical validations of the examined sensor are summarized in Table 1 and Table 2.

The developed sensor was applied to estimate acrylamide cations in foodstuff samples that were employed in general. The standard additions and the calibration curve techniques were employed. The statistical treatment of data provided that the calibration curve technique is better in the acrylamide estimation, while the standard additions technique is less acceptable. Thus, the error is not more than 1% because of the veracity, repeatability and reproducibility of the process. The mode of electrode construction was precise and accurate in comparison to the other previously listed techniques which are widely utilized in their estimation in foodstuff samples (Table 4).

In general, the quality of the produced data was excellent due to the optimum selection of the practical applications of the developed sensor. The consumed time in the determination is studied with no effect on the accuracy, and repeatability of the produced results.

## Figures and Tables

**Figure 1 membranes-11-00645-f001:**
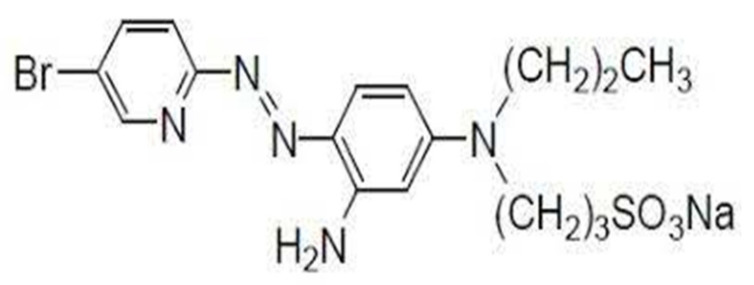
Structure of the reagent [BrPPSAA].

**Figure 2 membranes-11-00645-f002:**
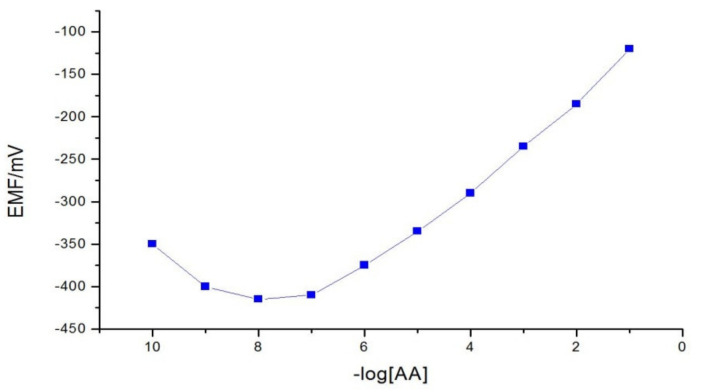
The suggested Acrylamide calibration curve in the linear concentration range 10^−7^–10^−1^ M.

**Figure 3 membranes-11-00645-f003:**
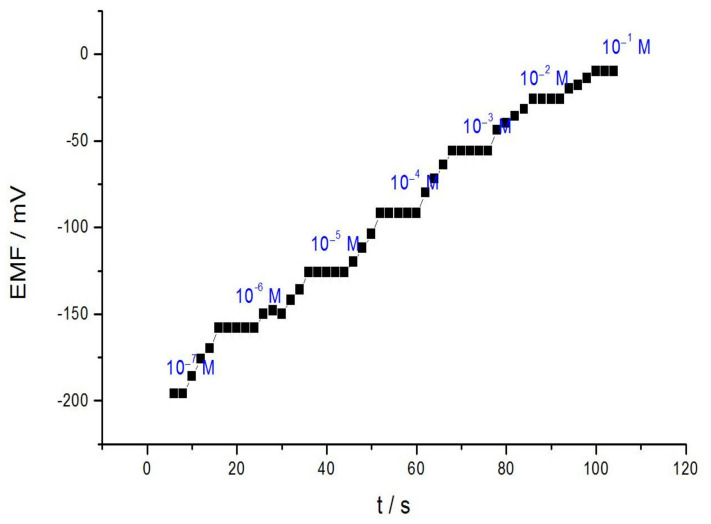
Response time of the suggested sensor for step changes in the concentration range of acrylamide.

**Figure 4 membranes-11-00645-f004:**
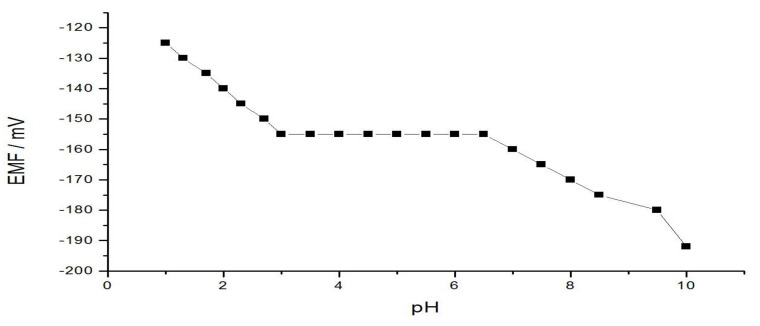
Effect of the electrode response on the pH in 10^−3^ M acrylamide solution.

**Table 1 membranes-11-00645-t001:** Analytical validation parameters of the developed acrylamide sensor.

Specific Slope, mV/decade	52.33
Intercept/mV	−46.40 ± 0.30
Limit of detection/mol dm^−3^	1.60 × 10^−8^
Linearity range/mol dm^−3^	1.0 × 10^−7^–1.0 × 10^−1^
Response time/s	8
Lifetime/d	120
pH working range	3.0–6.5

**Table 2 membranes-11-00645-t002:** The selectivity coefficients (K) values of AA membrane sensor (reference electrode Ag/AgCl).

K	E_i_ = E_j_	a_i_ = a_j_	MPM
Ca^2+^	0.211 ± 0.0031	0.272 ± 0.020	0.226 ± 0.0130
Mg^2+^	0.234 ± 0.0011	0.211 ± 0.002	0.215 ± 0.0310
Cu^2+^	0.274 ± 0.0032	0.268 ± 0.041	0.265 ± 0.0023
L-Alanine	-----	-----	0.046 ± 0.0031
Arginine	-----	-----	0.022 ± 0.0013
L-Cystine	-----	-----	0.024 ± 0.0023
Glucose	-----	-----	0.123 ± 0.0012
D-Fructose	-----	-----	0.156 ± 0.0041
Glycin	------	------	0.135 + 0.0032

**Table 3 membranes-11-00645-t003:** The recoveries of the added AA obtained by HPLC and by using the developed ion-selective electrode.

Foodstuff Sample	AA Added mg Kg^−1^	HPLC Technique	Calibration Curve Method			Standard Addition Method		
		Recovery ± SD ^a^	AA Found mg Kg^−1^	Relative Error %	Recovery ± SD ^a^	AA Found mg Kg^−1^	Relative Error %	Recovery ± SD ^a^
Roast Coffee (dry)	200	99.55 ± 0.25	199.25	0.75	99.63 ± 0.26	195.15	4.85	97.57 ± 0.35
Instant(soluble) coffee	350	99.65 ± 0.16	349.48	0.52	99.85 ± 0.32	346.03	3.97	98.86 ± 0.65
Potato chips	375	99.75 ± 0.23	374.45	0.55	99.85 ± 0.35	371.06	3.94	98.94 ± 0.54
Crisp bread	225	99.66 ± 0.34	224.55	0.45	99.80 ± 0.56	221.02	3.98	98.23 ± 0.72
French fries	120	99.45 ± 0.36	119.12	0.82	99.26 ± 0.45	116.15	3.85	96.79 ± 0.85
Toast	150	99.72 ± 0.55	149.54	0.46	99.69 ± 0.35	146.14	3.86	97.42 ± 0.66
Cearls based foods	175	99.80 ± 0.45	174.35	0.65	99.62 ± 0.31	171.25	3.75	97.85 ± 0.89

^a^ The averages of (five) estimations.

**Table 4 membranes-11-00645-t004:** Comparison of the interesting validation parameters of acrylamide [BrPPSAA] with some published selective electrodes for acrylamide determination.

Ref.	Linear Range (M)	Response Time (s)	Longlife (Day)	Detection Limit (M)	pH Range
This work data	1.0 × 10^−7^–1.0 × 10^−1^	8	120	1.6 × 10^−8^	3.0–6.5
33	7.0 × 10^−7^–7.0 × 10−^5^	------	9	3.9 × 10^−5^	7.4
34	5.0 × 10^−4^–7.0 × 10^−4^	<2	100	2.0 × 10^−3^	5.5
35	1.0 × 10^−5^–1.0 × 10^−4^	<2	120	1.0 × 10^−5^	5.0
36	1.0 × 10^−5^–2.0 × 10^−4^	10	------	3.0 × 10^−4^	-------
37	4.0 × 10^−5^–2.0 × 10^−4^	12	60	8.0 × 10^−9^	4.5
38	1.0 × 10^−8^–1.0 × 10^−5^	10	60	4.0 × 10^−8^	---------
39	5.0 × 10^−9^–7.5 × 10^−4^	20	-------	2.0 × 10^−8^	505

## Data Availability

The data that support the findings of this study are available on request from the corresponding author.

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
