# Peer review of "Potent Acrylamide Determination in Food Products Using Ion-Selective Electrode Technique"

_membranes, 2021, doi:10.3390/membranes11080645_

Round 1

Reviewer 1 Report

The developed electrode showed high performance and scientific soundness and originality is high.
So, I recommend to publish after minor revision.

In line 94, "the obtained" should be "The obtained".
In line 95, there is no period in sentence.
Totally, resolution of the figures must be improved.

Author Response

Answer to Reviewer No. 1

           Reviewer 1 said that " The developed electrode showed high performance and scientific soundness and originality is high. So, I recommend to publish after minor revision."

  1. In line 94 (old), becomes line 92 page 3 ,"the obtained" should be "The obtained". was corrected.
  2. In line 95 (old), there is no period in sentence., becomes line 93 page 3, recently published sensors
  3. The resolution of the figures was improved.
  4. All changes and modification are marked with red color in the attached revised marked form of the manuscript.

Reviewer 2 Report

A main drawback of the manuscript is its poor language level. Text need serious improvements not only from point of view of grammar, but also from point of view of formulations and style. This language level serious prevents clear understanding of presented information.

On the other hand the manuscript has also serious scientific problems.  I will mention only some of them:

  1. "chemical abstract number : 679787-08-7, pub chem ID: 329770275, molecular formula; C17H23BrN4Na2O6S, and molecular weight 537.34" -This information is not necessary to be given in a scientific paper.
  2. Line 75: "It forms an ion-associate complex with some different cations"
    The "some cations" must be exemplified.
  3. Lines 76-77: the molar absorptivity of those complex is 8.8 x 104 L mol-1 cm-1 which is considered a high value"
    All complexes have the same molar absorptivity? 
    Which is the relevance of the molar absorptivity of these complexes for the present study?
  4. Section 2.1 must also indicate the provenience of acrylamide and of the reagent BrPPSAA 
    On the other hand there are listed some reagents which do not appear in the manuscrpt to be used during the study (e.g. HF and H2SO4, TBP, tetrahydrofuran, methanol, asparagine) and other reagents used in the interference study (e.g. alanine, arginine , cystine, glycine, etc) are not listed in section 2.1.
  5. Line 107-108: "The cited reagent [BrPPSAA] was prepared and purified by means of crystallization with column chromatography as reported before [ 46 ]."
    Ref 46 does not describe the BrPPSAA preparation and purification. 
    Please check also the statement "purification by means of crystallization with column chromatography" of explain how does "crystallization with column chromatography" works?
  6. Line 120: "Stock solutions of Ca+2, Mg+2 and Cu+2 of 0.1 molar solutions were weighed"
    The solutions were weighed???
  7. Lines 117-118: "One gram of each sample ( 200, 350,375, 225, 120, 175 and 150 mg of them, respectively )"
    The amount of each sample was one gram or the mass given in brackets?
  8. The meanings of sections 2.5 and 2.6. are not clear for me. Ref 46 describes the formation of the complex between the BrPPSAA and cobalt. Which is the relevance of the BrPPSAA-cobalt complex for this study? Are there some similarities with the aa-BrPPSAA complex?
  9. Line 136: "An exactness weight of 0.01g active constituent [ AA ( BrPPSAA ) ]"
    How was the active constituent [AA(BrPPSAA)})] obtained?
    The authors should briefly explain in what conditions AA exists as a cation. 
  10. How was the membrane composition optimized?
  11. How was the limit of detection estimated?
    Figure 2 is not clear at all. According to figure 2 the points recorded for AA are overlapped by point given by other molecules.
    Which was the R or R2 of the calibration curve for AA determination?
  12. References should be given for equations (1) and (2). It should be also indicated what represents each parameter involved in the equations. 
    In Equation (2) appears only ai? ai stands for the activity of the interferent or of the analyte?
  13. "Solutions used for the determination of the response time for the tested sensor have these conditions: c1 : c2 = 1:100, v1 : v2 = 1: 20, where c1 is the concentration of the sample, c2 , the standard concentration, v1 is the sample volume and v2 1 is the volume of standard."
    What did the authors mean by sample solutions? Have they used the solutions obtained from the investigated food samples to asses the sensors' response time?
  14. I do not really understand Fig. 3. The response time for low concentrations (e.g. ~10 s for 10-7 M) is lower than the response time for higher AA concentrations (E.g. 110 s for 10-1 M)?
    On the other hand according to the legend of Figure 3 the graph shows the response time for a concentration of 10-3 M . 
  15. Line 189: "3.4. Influence of The Electrode Potential on the pH" and line 190 "The influence of the sensor potential on the pH"
    The pH influences the potential and not vice versa.

  16. Line 190-191: "was examined by recording the potential related to the chemical property of acrylamide" 
    Related to which chemical property of AA?
  17. Line 193: I do not understand what" the ratio of the electromotive force" means.
  18. 3.6. Determination of AA in Food Product Samples

    This section must be more concrete. The recoveries obtained for each investigated sample with the ISE and by HPLC must be given

    "The techniques of calibration curve, and standard additions were utilized. The amounts of AA in the samples were computed from predetermined calibration plots."
    Did the authors used the standard addition method or did they obtain the AA concentrations using the equation of the calibration plot?

The manuscript is not publishable in its present form.

Author Response

Answer to Reviewer No. 2

  • "Chemical abstract number : 679787-08-7, pub chem ID: 329770275, molecular formula; C17H23BrN4Na2O6S, and molecular weight 537.34" were canceled-This information is not necessary to be given in a scientific paper as stated by the Reviewer.

  1. Line 75 "It forms an ion-associate complex with some different cations"
    The "some cations" must be exemplified, cations are mentioned at lines74-75.
  2. The provenience of acrylamide and of the reagent BrPPSAA was stated at Section 2.1 page 3.
  3. At Section 2.1. page 3, the listed reagents used during the study are revised.
  4. "The cited reagent [BrPPSAA] was prepared as reported before [ 46 ]."
    was checked and the reference no. becomes [ 49 ].
  5. "Stock solutions of Ca+2, Mg+2 and Cu+2 of 0.1 molar solutions were weighed", the statement was checked and revised at page 3 lines 107-108.
  6. "One gram of each sample ( 200, 350,375, 225, 120, 175 and 150 mg of them, respectively )", The sample preparation section was revised at page 3 lines 114-119.
  7. Sections 2.5 and 2.6 becomes Sections 2.4 and 2.5 after revision and modification, they have been reviewed and clarified at pages 3 & 4 lines 124-133 & lines 135-144, respectively.
  8. Preparation of the active constituent [AA(BrPPSAA)] was stated at Section 2.5 at page 3 lines 135-140.
  9. The membrane composition was explained at Section 2.6 page 4 lines 146-156.
  10. Reference no. [ 52 ] was given for the calculation of the selectivity coefficient.
  11. Section 3.3. Dynamic Response Time of the Acrylamide Sensor was revised and clarified at page 6 lines 192-199.
  12. The practical response time was registered by changing the acrylamide cations concentration in solution, over step changes in the concentration range from 1.0×10−7-1.0×10−1 M, and as presented in Fig. 3, the acrylamide sensor reached its equilibrium response over the whole concentration range, in a very short time (8 s) , and the legend of Figure 3 was changed.
  13. "The influence of the sensor potential on the pH" was corrected at page 6 line 201.
  14. "was examined by recording the potential related to the chemical property of acrylamide"  was examined at page 6 lines 203.
  15. The ratio of the electromotive force; EMF of the AA sensor system with respect to reference electrode was read after the stability of the sensor’s response. This is the meaning of the ratio., at page 6 line 203.
  16. Table 3. Acrylamide (AA) content in foodstuffs as estimated by the developed  membrane compared to HPLC technique was re-tabulated to be more clarified.

  1. All changes and modification are marked with red color in the attached revised marked form of the manuscript.

Reviewer 3 Report

The present work reports the realization of a liquid potentiometric membrane sensor for acrylamide (AA) determination, based on the reagent 2-(5-Bromo-2-pyridylazo)-5-[N-n-Propyl-N-(3-Sulfopropyl)amino]aniline [BrPPSAA]. The authors studied the sensitivity, selectivity and reliability of the sensor. The sensor was employed for the determination of AA in a series of food products and a comparison with High Pressure Liquid Chromatography (HPLC) technique was also performed. The obtained results show that the developed sensor could be employed for AA determinations in foodstuff samples.

The results look reasonable and support the observed trends, the present work could be publishable on Membranes only after some minor revisions.

Suggested remarks:

Figures 2, 3, 4: it is suggested a better resolution.

A deep editing of English is needed for a better comprehension of the readers.

The acronyms present in the text (particularly in the abstract and in the introduction) could be firstly explained to be understandable to Non-specialists.

More recent works dealing with the application of electrochemical techniques in biological analysis should be considered by the authors, i.e https://doi.org/10.1039/D0NJ03896B, https://doi.org/10.1021/acsbiomaterials.9b01659, etc

Author Response

Answer to Reviewer No. 3

Reviewer 3 said that: The results look reasonable and support the observed trends,

   the present work could be publishable on Membranes only after some minor revisions.

- The introduction provide sufficient background and include all relevant references,

- The research design is appropriate, the methods are adequately described, the  results are clearly presented, and the conclusions are supported by the results.

1.    The resolution of the figures was improved.

2. All the acronyms present in the Abstract, Introduction and the text were explained

     to be understand to non-specialists.

3. The whole manuscript was re-checked with respect to grammar and statements.

4. The Introduction was revised and three References numbers [ 22-24 ]have been added to support and complement the manuscript.

5. All changes and modification are marked with red color in the attached revised marked form of the manuscript.

Reviewer 4 Report

Authors presented a potent selective acrylamide liquid sensor based on the reaction of acrylamide with 2-(5-Bromo-2-pyridylazo)-5-[N-n-Propyl-N-(3-Sulfopropyl) amino] aniline.  The sensor shows analytical activity based on the reaction between BrPPSAA and analyte. Authors reported that the acrylamide and BrPPSAA formed ion-associate complex in a polyvinyl chloride (PVC). The authors did not provide any experimental evidence for the interaction of the analyte with BrPPSAA. NMR, IR or Mass spectrometry study for complexes was be added.

The results presented in this work are interesting.  For many years, analogic compound  2- (5-Bromo-2-pyridylazo) -5- [N-propyl-N- (3-sulfopropyl) amino] phenol disodium salt dihydrate is used for spectrophotometric detmination of Zn (II), Cu (II), Fe (II), Co (II) and H2O2.  BrPPSAA react with metal cations, also. Are metal cations interferents in the determination of acrylamide? How is the measurement of AA in the presence of metal cations? The paper presented measurement of standard solutions of the metal cation itself and AA alone?

The construction of a selective electrode arouses the interest of chemists each time; please add a photo and scheme of the electrode construction and the entire measuring system to the publication.

Author Response

Answer to Reviewer No. 4

               Reviewer 4 said that "The results presented in this work are interesting."

  1. Some characterizations of the cited reagent and the interaction between the acrylamide and the reagent were provided at page 2 lines 71-79 and at Section 2.5. pages 3 & 4 lines 135-144.
  2. The measurement of Acrylamide in the presence of metal cations, the cited reagent BrPPSAA react with metal cations, the interference can be eliminated , this was explained at " The Discussion" page 8 lines 260 - 273.
  3. All changes and modification are marked with red color in the attached revised marked form of the manuscript.

Round 2

Reviewer 2 Report

I have read the revised form of the manuscript with the title "Potent Acrylamide Determination in Food Products Using Ion-Selective Electrode Technique". The authors responded satisfactory to some of my previous comments and suggestions but some of them are still unsolved. I make again some suggestions, that I hope to help improving both the scientific content and the English language of the manuscript:

  1. Lines 29-30: "acrylamide causes cancer in 30 the diet in humans and animals." Please revise the sentence because AA can not cause cancer in the diet, it can cause cancer in humans.
  2. Lines 128-129: "The cited reagent [BrPPSAA] is the active membrane constituent. The complex, plasticizer, and the polyvinyl chloride were grounded, then adding tetra hydro furan as a volatile solvent"
    a. The active component of the membrane is BrPPSAA or its complex with AA?
    b. If the complex of BrPPSAA with AA is the active component the authors must indicate how was it obtained. I asked this in mys previous review and the authors respond:"Preparation of the active constituent [AA(BrPPSAA)] was stated at Section 2.5 at page 3 lines 135-140." Unfortunately at this lines there is no working procedure for the "synthesis" of the [AA(BrPPSAA)] . There are some information about theoretica aspects and about the absorption spectrum.
    c. The compounds used as "complex and the plasticizer" must be named.
  3. Line 130:"an appropriate diameter disk was cut and glued"
    Where was the cut disk glued? Please give some more concrete details.
  4. Lines 136-137:"It performs ion-associate complexes  with any cations." 
    Which is the significance of this statement for the present study?
  5. "The maximum absorption of free 5-Br-PSAA is at 457 nm. The UV-visible absorption spectrum of the acrylamide-5-Br-PSAA ion-associate complex was measured in the range of 300 – 700 nm at the optimum acidic pH. A maximum absorbance of the complex is observed approximately at 602 nm where the free reagent has minimum absorbance."
    As this paper does not deal with the spectrometric investigation of the 5-Br-PSAA and acrylamide-5-Br-PSAA UV-VIS absorption  behavior, the authors must explain which is the relevance of this fragment for the present study. 
  6. Lines 150-153:"From the calibration curve( Fig. 2 ) the best behavior of the membrane electrode consists of an exactness weight of 0.01g active constituent [ Acrylamide( BrPPSAA ) ] with a blend of, 0.44 g PVC , and 0.55 g TEHP were blended to provide the ion-selective
    sensor strata."
    a. Please revise the sentence because it may lead to miss-understandings.
    b. Fig. 2 doesn't show that the best performance of the ISE is for the indicated composition. 
    c. Did the authors tested also other compositions? In my previous report I have asked "How was the membrane composition optimized?" and the authors responded: "The membrane composition was explained at Section 2.6 page 4 lines 146-156."
    At these lines there is no information regarding the optimization of the membrane composition. Why did they select this composition and not another  for the membrane preparation?
  7. Lines 170-171: "Figure 2 presented the acrylamide sensor’s calibration curves detected in acrylamide (AA), and its interfering metal cations, and compounds of 10-3 molar solutions."
    I suggest to formulate this sentence as:"Figure 2 presented the acrylamide sensor’s calibration curves detected in acrylamide (AA) in the absence and in the presence of 10-3 M of possible interfering species.
  8. Line 174:"presented the analytical validations of the suggested acrylamide sensor."
    Formulate as "presented the analytical validation parameters of the suggested acrylamide sensor."
  9. The authors did not respond to my previous questions: 
    a. How was the limit of detection estimated?
    b. Which was the R or R2 of the calibration curve for AA determination?
  10. Figure 2. What means pa on the Ox axis? pa must be defined.
  11. Title of Table 1:"Analytical validations parameters of the developed acrylamide sensor matrix.(reference electrode Ag/AgCl) membrane sensor preparation."
    I do not understand what the authors mean by "of the developed acrylamide sensor matrix.(reference electrode Ag/AgCl) membrane sensor preparation."
    I suggest ""Analytical validation parameters of the developed acrylamide sensor"
  12. Table 1. "Specific Slope / mV" - The units for the slope are not mV, but mV/decade.
  13. Please use during the whole manuscript the same number of decimal places  e.g. Table 1:  57.60 and -46.30 instead of 4.3. 
  14. The authors did not respond to my previous remarks:
    a. It should be also indicated what represents each parameter involved in the equations (1) and (2). 
    b. In Equation (2) appears only ai? ai stands for the activity of the interferent or of the analyte? 
    Please check both equations because they contain errors!
  15. Lines 202-203: "The dependence of pH on the sensor potential was examined by recording the potential related to the chemical property of acrylamide as cations in an acidic medium
    203".
    a. As I have mentioned in my previous review report the sensor potential depends on the pH and not the pH depends on the potential.
    b. The potential was recorded related to which chemical property of AA?
    AA  is involved in acid-base equilibrium and thus the concentration of the cationic form of AA changes with pH. resulting in changes in the sensor potential. 
  16. As I have mentioned in my previous report;
    Line 206:  I do not understand what" the ratio of the electromotive force" means.
  17. Line 207: "after the stability of the sensor’s response" change to "after the stabilization of the sensor’s response"
  18. Lines 208-209: Please revise :"Lowering and higher than the working pH range ( 3.0-6.5 ), at higher pH values"
  19. Line 210: Please revise the formulation: "the incompleteness of the ion-associate complex"
    The incompleteness can not be of the complex it may be of the complexation reaction.
  20. please be consequent in writing . e.g mV decade-1 or  mV/decade; M not molar
  21. Line 243: "the Nernstian value was 57.6 mV/decade as presented in Fig. 2," 
    a. The Nernstian value is always 59.15 mV/decade at 25oC. 57.6 mV/decade is the slope of the sensor described in this paper
    b. The slope is not presented in Fig. 2. Fog. 2 presents the variation of the EMF with -log[AA} without showing the calibration curve and its regression equation with the slope representing the sensor response.
  22. Lines 244-245:"It means any increment in the concentration of 10-3 M test solutions, the potential change of 57.6 mV/decade."
    This statement is not correct. Please reformulate e.g. as: "It means that the potential changes with 57.6 mV if the AA concentration changes with one order of magnitude."
  23. Line  247: Change "57.6 mV" to "57.6 mV/decade"
  24. Lines 248-249:"Knowing that, the composition of the membrane and the nature of additives used to affect the selectivity and sensitivity of the ion-selective electrode." 
    I do not understand what the authors mean with this sentence. Please revise it.
  25. Lines 249-254 are identical with lines 146-153.
  26. Lines 254-257: "The electrode showed a
    super Nernstian response with a slope of 57.60 mV per decade for four replicate measurements in the linearity range of 1.0×10−7-1.0×10−1 M in standard solutions  of acrylamide cations 10-3 M as seen in Figure 3."
    a. Figure 3 shows the response time.
    b. "in the linearity range of 1.0×10−7-1.0×10−1 M in standard solutions  of acrylamide cations 10-3 M " . Please revise. I do not understand what did the authors mean by " the linear range of 1.0×10−7-1.0×10−1 M in standard solutions  of acrylamide cations 10-3 M". Why is the acrylamide cations concentration 10-3 M?
  27. Line 261-263: "Table 2 revealed that the suggested acrylamide ion-selective electrode is highly selective towards acrylamide cations satisfactorily in the presence of those foreign metal cations, and compounds under test."
    The selectivity of this electrode is "highly selective towards acrylamide cations" or is it "satisfactorily"?
  28. Lines 262-265:"ion-selective electrode is highly selective towards acrylamide cations satisfactorily in the presence of those foreign metal cations, and compounds under test. As it was shown from the resulting data, the investigated interfering metal cations ( Ca+2, Mg+2, and Cu+2 ) may be interfering."

    There is a contradiction: "ion-selective electrode is highly selective towards acrylamide cations satisfactorily in the presence of those foreign metal cations, " and " the investigated interfering metal cations ( Ca+2, Mg+2, and Cu+2 ) may be interfering."
    The authors must clearly indicate if the electrode is selective in the presence of the metal cations or may this metal cations interfere.

  29. Table 2. Why are the selectivity coefficients given as a sum of two values?

  30. Title of Table 2. "... acrylamide (AA) electrode matrix.(reference electrode  Ag/AgCl) membrane sensor preparation."
    a. it is enough to write AA because the abbreviation was already explained'
    b. the selectivity coefficients are of the electrodes and not of the electrode ...preparation. 
    c. it is enough to write electrode or sensors.
    Please correct as: "AA membrane electrode (reference electrode Ag/AgCl)".

  31. From Table 2 I do not understand which is the significance of the first line, regarding AA. Did the authors estimate the selectivity coefficient of the sensor towards AA? But AA (acrylamide) is the analyte not the interferent.

  32. Table 3. The title is not correct because the table does not indicate the AA content in the analyzed samples, but the recoveries of the added AA obtained by HPLC and by using the developed ion-selective electrode.
  33. Table 3. HPLC is a technique but "Calibration curve" and "standard addition" are not techniques, they are methods. Please correct "Calibration curve technique" and "Standard Addition Technique" with "Calibration curve method" and "Standard addition method"
  34. Lines 302-304: "As can be seen in Table
    4 a comparison among some of the interesting validations of the quantitative estimations..."
    Table 4 contains a comparison of the "validation parameters" not of the "validations"
  35. Line 330; Please replace "the mistake" with "the error".
  36. Lines 335-336: this statement "the quality of the produced data was excellent due to the good luck in selected practical applications applying our suggested fabricated electrode" seems very strange to me.
    The obtained data are due to the good luck???? I suppose the authors mean "due to the optimum selection of the practical applications of the developed sensor" 
  37. The whole document must be carefully checked in order to avoid misunderstandings.
  38. English language must be improved.

Author Response

Editor- in-Chief

        Membranes

     Dear Editor,

             Thank you for your e-mail concerning the manuscript of ID reference: Membranes-1307897, entitled: “Potent Acrylamide Determination in Food Products Using Ion-Selective Electrode Technique ”                                                                                                         

Reply to Academic Editor

            I have studied well the other comments raised by the Academic Editor and the Reviewer No. 2 and all the comments were considered, the paper has been revised again for the second time. The amendments and modifications done are in the following list:-

  1. Lines 124-129, "The selective membrane fabrication was developed as mentioned earlier [ 51 ]. It contains a column electrode of Teflon mutable and a body filled with a membrane liquid phase " + Ag/Ag Cl " an internal reference sensor/1.0×10-2 mol L−1 acrylamide solution, 1.0×10-2 mol L−1 KCl (internal reference solution). The cited reagent [BrPPSAA] is the active membrane constituent. The complex, plasticizer, and the polyvinyl chloride were grounded, then adding tetra hydro furan as a volatile solvent". were revised, and the names of the complex and the plasticizer were added.
  2. Lines 129-131 and 164-165 are in conflict, were revised.
  3. Lines 144-157: several repetitions were revised.
  4. Line 257. Figure capture: was corrected, Figure 2 instead of Figure 3.
  5. Lines 259-260, the slope of 57.6 mV/decade is called “super Nernstian” was changed to a good because it is near from the real value 59.1 mV/decade.
  6. Ref. 24 was added, to support the references of electrochemical techniques.
  7. Terminology were corrected using well established terms.
  8. The term “strata” in this context means " layer ", it was revised.
  9. “electrode was acclimatized” was revised to “electrode was conditioned”.
  10. The whole manuscript was re-checked with respect to grammar and statements to improve English language.
  11. Table 2 was revised and In case of sugars and amino acids, ( Matched potential method is employed which depending on the ratio of the ai and aj ( activities of the acrylamide and the interfering compounds, respectively )the high selectivity is in relation to the difference in polarity and lipophilic nature of their molecules relative to acrylamide. The mechanism of selectivity is basically based on the stereo-specificity and electrostatic environment, and counts on how much fitting is available between the locations of the lipophilicity sites in two competing species in the bathing solution side and those present in the receptor of the ion-exchanger, Reference no. [54] was added to support the claims.
  12.     All changes and modification are marked with red color in the attached revised marked form of the manuscript.

         Thanking you in anticipation, please accept my best regards.

                                                                                Yours Sincerely,

                                                                             Prof. Dr. Sabry Khalil

Answer to Reviewer No. 2

            Many thanks for Reviewer 2 , the manuscript was improved by studying and revising based on his comments which are listed as follow:-

  1. Lines 29-30: "acrylamide causes cancer in line 30 the diet in humans and animals" was revised.
  2. Lines 127-128 The active component of the membrane is BrPPSAA
  3. Line 128 The compounds used as "complex and the plasticizer" are named.
  4. "an appropriate diameter disk was cut and glued" Where was the cut disk glued? was stated at Line 130-131.
  5. Lines 136-137:"It performs ion-associate complexes  with many cations." Which is the significance of this statement for the present study? This statement was canceled.
  6. "The maximum absorption of free 5-Br-PSAA is at 457 nm. The UV-visible absorption spectrum of the acrylamide-5-Br-PSAA ion-associate complex was measured in the range of 300 – 700 nm at the optimum acidic pH. A maximum absorbance of the complex is observed approximately at 602 nm where the free reagent has minimum absorbance."
    As this paper does not deal with the spectrometric investigation of the 5-Br-PSAA and acrylamide-5-Br-PSAA UV-VIS absorption  behavior, the authors must explain which is the relevance of this fragment for the present study. To illustrate the formation of a complex bet. reagent and AA.
  7. Lines 150-153:"From the calibration curve( Fig. 2 ) the best behavior of the membrane electrode consists of an exactness weight of 0.01g active constituent [ Acrylamide( BrPPSAA ) ] with a blend of, 0.44 g PVC , and 0.55 g TEHP were blended to provide the ion-selective sensor strata." The statement was revised.
  8. . Fig. 2 doesn't show that the best performance of the ISE is for the indicated composition. From Fig. 2 we get the value of the characteristic slope.
  9. "How was the membrane composition optimized?" The membrane composition was studied as explained at Section 2.6 page 4 lines 146-153." the statements were revised and modified.
  10. Lines 170-171: "Figure 2 presented the acrylamide sensor’s calibration curves detected in acrylamide (AA), and its interfering metal cations, and compounds of 10-3 molar solutions." this sentence formulate to:"Figure 2 presented the acrylamide sensor’s calibration curves detected in acrylamide (AA) in the absence and in the presence of 10-3 M of possible interfering species.
  11. Line 175:"presented the analytical validations of the suggested acrylamide sensor." this sentence Formulate to "presented the analytical validation parameters of the suggested acrylamide sensor."
  1. Figure 2. What means pa on the x axis? pa was defined under the Fig.
  2. Title of Table 1 was changed to "Analytical validation parameters of the  developed acrylamide sensor".
  3. Table 1. "Specific Slope / mV" - The units for the slope are not mV, but mV/decade. was revised.
  1. During the whole manuscript the same number of decimal places  e.g. Table 1:  57.60 and -46.30 instead of 4.3.  was revised.
  1. Each parameter involved in the equations (1) and (2) was defined. and the both equations are checked. 
  2. Lines 207-209 : the sensor potential depends on the pH and not the pH depends on the potential. was revised. and "the sensor potential was examined by recording the potential related to the chemical property of acrylamide as cations in an acidic medium" was revised.
  1. Line 212:  " the ratio of the electromotive force" was revised.
  2. Line 213: "after the stability of the sensor’s response" change to "after the stabilization of the sensor’s response" was corrected.
  3. Lines 214-218: Please revise :"Lowering and higher than the working pH range ( 3.0-6.5 ), at higher pH values" was revised.
  1. The estimation of the limit of detection was stated at line 173 , and the value of R2 was stated at line 174.
  1. Line 216: the formulation: "the incompleteness of the ion-associate complex" was revised to "the incompleteness of the complexation reaction."
  1. The consequent in writing . e.g mV decade-1 or  mV/decade; M not molar was revised.
  1. Line 248: "the Nernstian value was 57.6 mV/decade as presented in Fig. 2," This value was obtained from the slope of Fig. 2.
  1. Lines 249-250:"It means any increment in the concentration of 10-3 M test solutions, the potential change of 57.6 mV/decade."
    This statement is reformulate to: "It means that the potential changes with 57.60 mV/decade if the AA concentration changes with one order of magnitude."
  2. "57.6 mV" was changed to "57.60 mV/decade" through the whole manuscript.
  1. Lines 253-254:"Knowing that, the composition of the membrane and the nature of additives used to affect the selectivity and sensitivity of the ion-selective electrode."  was revised to be clear.
  1. Lines 249-254 are identical with lines 146-153. was rechecked.
  2. Line 257   "as seen in Figure 3" was checked Figure 2.
  1. Line 261 "in the linearity range of 1.0×10−7-1.0×10−1 M in standard solutions  of acrylamide cations 10-3 M " was revised.
  1. Line 267 :"The selectivity of this electrode is "highly selective towards acrylamide cations " was revised.
  2. The electrode is selective in the presence of the metal cations or may this metal cations interfere " , This was explained at lines 268-274.
  1. Title of Table 2 was changed to : "The values of selectivity coefficients ( K ) of AA membrane electrode (reference electrode Ag/AgCl)".
  1. The validation of analytical procedure( R1 ) for the determination of acrylamide with respect to repeatability and reproducibility was discussed at page 9 lines 300-311.
  1. Table 2 , the first line, regarding AA was canceled.
  1. The title of Table 3 was corrected.
  1. Table 3 was corrected with respect to "Calibration curve method" and "Standard addition method", instead of techniques.
  1. Line 314: "As can be seen in Table 4 a comparison among some of the interesting validations of the quantitative estimations..." changed into "validation parameters".
  1. Line 341; "the mistake" was replaced with "the error".
  1. Lines 346-347: The statement "the quality of the produced data was excellent due to the good luck in selected practical applications applying our suggested fabricated electrode" was revised and corrected to "due to the optimum selection of the practical applications of the developed sensor". 
  2. The whole manuscript was carefully checked to avoid misunderstandings.
  1. The whole manuscript was re-checked with respect to grammar and statements to improve English language.         Thanking you in anticipation, please accept my best regards.                                                                             Prof. Dr. Sabry Khalil
  2.                                                                                 Yours Sincerely,
  3.      43.All changes and modification are marked with red color in the attached revised marked form of the manuscript.

Round 3

Reviewer 2 Report

I have carefully read  the second revised form of the manuscript and it was improved but there are still some unsolved important aspects, some of them being reported since my first review.

  1. Lines 15-17: The sensor was successfully
    utilized to estimate acrylamide in food product samples. The deployed sensor was successfully applied for rapid inexpensive analysis of acrylamide cations in food products..." 
    In my opinion these two sentences must be merged because they contain almost the same information.
  2. Lines 26-27: "acrylamide is formed at the boiling time/sterilizing with foods containing starch, like bread and potatoes are cooked" 
    The sentence must be revised. I recommend ""acrylamide is formed during the boiling/sterilizing of foods containing starch, like bread and potatoes."
  3. The same observations as in my previous two reviews:
    a. Lines 127-128: "The cited reagent [BrPPSAA] is the active membrane constituent. The complex; {Acrylamide[BrPPSAA]},"
    In my opinion it is a contradiction when the authors wrote that the "active membrane constituent is [BrPPSAA} and it does not appear in the membrane composition, but instead the membrane contains the complex of this reagent with AA,.
    b. The  working procedure for the "synthesis" of the [AA(BrPPSAA)] must be described.
  4. Line 142: "optimum acidic pH". The exact value of the optimum pH must be given and to demonstrate that the  absorption maximum at 602 nm belongs to the complex the authors should also indicate the absorption maximum of acrylamide, not only of the [BrPPSAA].
  5. In my previous two reports I asked  how did the authors optimized the membrane composition and if they have tested also other compositions?"
    The responses given by the authors did not give any information related to the exact membrane composition optimization.
    The las response was similar to the previous one, namely: "The membrane
    composition was studied as explained at Section 2.6 page 4 lines 146-153"
    Unfortunately, reading these lines  I didn't find any information related to my question.
    Lines 147-149: "Therefore, the influence of various membrane compositions on the response characteristics of the fabricated acrylamide membrane electrode was studied to detect the best composition of the nominated electrode."
    The authors should indicate  which were the tested compositions and how did they select that one which was used for the membrane in this study?
  6. Lines 150-152: "Thus, From the characteristic slope ( 57.60 mV/decade )of the calibration curve(Fig. 2 ) the best behavior of the membrane electrode consisting of an exactness weight of 0.01g of the compound[ Acrylamide( BrPPSAA ) ] with 0.44 g PVC , and 0.55 g TEHP which were blended to provide the ion-selective sensor layer"
    a. The author have revised the sentence but, in my opinion, the revised form is also not grammatically  corrected.
    b. The slope value of the calibration curve of 57.60 mV/decade cannot be seen in Figure 2. A regression equation is necessary to emphasize this value.
  7. Lines 172-173: "the limit of detection is 1.6 x10 -8 M as determined from the intersection of the two extrapolated segments of the calibration graph,"
    I do not understand how was the limit of detection estimated. Which are the two segments of the calibration graph which were extrapolated?
  8. Legend of Fig 2 is not clear for me. Does it represent the sensor response to AA concentrations in the presence of 10-3 M of each indicated possible interfering species? If yes, than I recommend to formulate as 
    "Acrylamide;( AA ) sensor calibration curves in the absence,( K ) and in the presence of 10-3 M  of possible interfering species  ,( J )Ca+2,( I ) Mg+2,( H )Cu+2,( G ) L-Alanine,( F ) Arginine,( E ) L-Cystein ,(D)Glucose, (C ) D-Fructose, and ( B )Glycin ."
  9. log Kpot ij = (Ej - Ei) / S - ( Zi / Zj - 1 ) log ai
    is the correct form of equation (1)
    and  in Eq (2) the ions charges are missing now.
    What represents the subscript M in eq (2)?
  10. "Ei is the potential of AA, Ej is the potential of the interfering ion,"
    formulate as:
    "Ei is the potential of AA solution, Ej is the potential of the interfering ion solution,"
  11. This sentence "By using the separate solution method, at the value of EMF with acrylamide cations concentration 0.001 M and, the potential –160 mV" must be revised
    I suppose that the authors mean that they used a 10-3 M concentration of analyte and interferent, respectively, for the separate solution method and the potential of -160 mV when they applied the matched potential method.
  12. Line 214 "Lowering and higher than the working pH range ( 3.0-6.5 ),"
    I do not understand this fragment. Revise or delete it.
  13. Lines 217-218: "At lower pH values the potential increases (-150 at pH 2.5, and -157 at pH 2.0) attributed to the membrane responses to hydronium ions and acrylamide cations.," formulate as " At lower pH values the potential increases (-150 at pH 2.5, and -157 at pH 2.0) due to the membrane responses to hydronium ions and acrylamide cations."
  14. Legend Figure 4. change to "Figure 4. Effect of pH on the sensor response  in 10-3 M acrylamide solution'
  15. Lines 247-248. "In this research, the Nernstian value was 57.6 mV/decade as presented in Fig. 2, which is closely agreed with the ideal value."
    As I have indicated in my previous report 
    a. The Nernstian value is always 59.16 mV/decade at 25oC. 57.6 mV/decade is the slope of the sensor described in this paper
    b. The slope is not presented in Fig. 2. Fig. 2 presents the variation of the EMF with -log[AA} without showing the calibration curve and its regression equation with the slope representing the sensor response.
  16. "Knowing that the composition of the membrane and the nature of additives used affect the selectivity and sensitivity of the ion-selective electrode."  is grammatically not correct. Formulate it as :"It is known that the composition of the membrane and the nature of additives used affect the selectivity and sensitivity of the ion-selective electrode."
  17. Lines 254-257: "Thus, the effect of different membrane compositions on the response characteristics of the constructed acrylamide ion-selective electrode was examined to detect the best composition of the proposed electrode."
    As I have previously mentioned, the authors did not discuss the effect of membrane compositions. To test the effect of membrane compositions on the electrode response means to prepare several membranes with different compositions and to discuss their response toward the analyte. 
  18. I have already mentioned in my previous review reports and also earlier in this review Fig. 2 does not emphasize that the given membrane composition  has "the best" performances. To state that one membrane is "the best" , it must be compared to other membranes. 
  19. Line 261: ""in the linearity range of 1.0×10−7-1.0×10−1 M in standard solutions  of acrylamide cations 10-3 M " . 
    As I have also written in my previous report:
    Please revise. I do not understand what did the authors mean by " the linear range of 1.0×10−7-1.0×10−1 M in standard solutions  of acrylamide cations 10-3 M". Why is the acrylamide cations concentration 10-3 M?"
    The authors responded that they have revised the sentence but in the last version of the manuscript, downloaded from the journal site, it is unchanged.
  20. Lines 262-263: "As seen in Figure 2 the developed electrode presented constant potential readings for day-to-day measurements"
    It cannot be seen from Fig. 2  that the electrode has constant potential readings for day-to-day measurements because Fig. 2 does not show the readings or the calibration graphs obtained on different days.
  21. replace "molar" with "M" (abstract, sections 2.2 and 2.4).
  22. The authors did not respond to my previous comment: "Table 2. Why are the selectivity coefficients given as a sum of two values?"
  23. Lines 294-295:"Also, as can be seen from the provided data ( Table 3 ) that the calibration curve and the standard additions techniques were utilized"
    Despite the fact that this sentence brings no new information and can be deleted, it is not correct. Please formulate it as: "Also, as can be seen from the provided data ( Table 3 )  the calibration curve and
    the standard additions methods were utilized."
    Line 296: replace "the technique of the calibration curve" with "the method of the calibration curve"
    Line 297: replace "the standard additions technique" with "the standard additions method"
  24. Please replace "interesting validation parameters" with "important validation parameters" (e.g. line 314, line 325)
  25. The whole document must be carefully checked in order to avoid misunderstandings. English language must be further improved.

Author Response

To Editor- in-Chief         on Fri.,13 Aug.,2021

        Membranes

     Dear Editor,

             Thank you for your e-mail concerning the manuscript of ID reference: Membranes-1307897, entitled: “Potent Acrylamide Determination in Food Products Using Ion-Selective Electrode Technique ”                                                                                                         

Reply to Academic Editor

            I have studied well the other comments raised by the Reviewer No. 2 for the third time and all the comments were considered, the paper has been revised again. The amendments and modifications done are in the following list:-

Answer to Reviewer No. 2

            Many thanks for Reviewer 2 , I think the manuscript is well improved by studying and revising based on his comments which are listed as follow:-

  1. Lines 15-17: The sensor was successfully utilized to estimate acrylamide in food product samples. The deployed sensor was successfully applied for rapid inexpensive analysis of acrylamide cations in food products..." was revised
  2. Lines 25-26: "acrylamide is formed at the boiling time/sterilizing with foods containing starch, like bread and potatoes are cooked" The sentence must be revised. I recommend ""acrylamide is formed during the boiling/sterilizing of foods containing starch, like bread and potatoes." was revised
  3. Lines 128-129"The cited reagent [BrPPSAA] is the active membrane constituent. The complex; {Acrylamide[BrPPSAA]}," was revised
  4. " The exact value of the optimum pH , the  absorption maximum at 602 nm belongs to the complex , the absorption maximum of free acrylamide, and the cited reagent [BrPPSAA]. All are indicated at lines 136-146.
  5. The optimization of the membrane composition was described at Section 2.6 lines 148-162.
  6. Figure 2 was re-plotted to see the linear range of the membrane electrode, the Nernstian value of the calibration curve 52.33 mV/decade was calculated from the slope ( Fig. 2 ).
  7. Lines 178-179: "the limit of detection is 1.6 x10 -8 M as determined from the intersection of the two extrapolated segments of the calibration graph ( Fig. 2 )"

  8. Legend of Fig 2 was corrected.
  9. The subscript M in Eq (2) represents the interfering ion or compound (j), it was stated at line 202.
  10. "Ei is the potential of AA solution, Ej is the potential of the interfering ion solution," was corrected at lines 196-197.
  11. The sentence "By using the separate solution method, at the value of EMF with acrylamide cations concentration 0.001 M and, the potential –160 mV" was revised at lines 198-200.
  12. Line 222-223 "Lowering and higher than the working pH range ( 3.0-6.5 ) was revised.
  13. Lines 225-227: "At lower pH values the potential increases (-150 at pH 2.5, and -157 at pH 2.0) attributed to the membrane responses to hydronium ions and acrylamide cations.," was corrected.
  14. Legend Figure 4. change to "Figure 4. Effect of pH on the sensor response  in 10-3 M acrylamide solution"
  15. Lines 256-257. "In this research, the Nernstian value was 57.6 mV/decade as presented in Fig. 2, which is closely agreed with the ideal value." was revised.

  16. The effect of membrane compositions on the electrode response was discuss at Lines 262-267.
  17. Line 266: ""in the linearity range of 1.0×10−7-1.0×10−1 M in standard solutions  of acrylamide cations 10-3 M " was revised. 

  18. Lines 262-263 (Old): "As seen in Figure 2 the developed electrode presented constant potential readings for day-to-day measurements" was canceled
  19. "molar" was replaced with "M" at (abstract, sections 2.2 and 2.4).
  20. Lines 300-301:"Also, as can be seen from the provided data ( Table 3 ) that the calibration curve and the standard additions techniques were utilized"
    formulated as: "Also, as can be seen from the provided data ( Table 3 )  the calibration curve and the standard additions methods were utilized."
  21. Line 302: replace "the technique of the calibration curve" with "the method of the calibration curve", was revised.
    - Line 303: replace "the standard additions technique" with "the standard additions method", was revised.
  22. "interesting validation parameters" was replaced with "important validation parameters" (e.g. line 320, line 331)
  23. The whole manuscript was carefully checked to avoid misunderstandings.     25.All changes and modification are marked with red color in the attached revised marked form of the manuscript.                                                                                Yours Sincerely, 
  24.                                                                              Prof. Dr. Sabry Khalil
  25.          Thanking you in anticipation, please accept my best regards.
  26. 24.The whole manuscript was re-checked with respect to grammar and statements to improve English language.